# Evaluation of Real-Time Perception of Deformation State of Host Rocks in Coal Mine Roadways in Dusty Environment

Pengfei Shan [1,2], Chengwei Yan [1,2,*], Xingping Lai [1,2], Haoqiang Sun [1,2], Chao Li [2,3] and Xingzhou Chen [2]

1 School of Energy Engineering, Xi'an University of Science and Technology, Xi'an 710054, China
2 Key Laboratory of Western Mines and Hazard Prevention of Ministry of Education, Xi'an University of Science and Technology, Xi'an 710054, China
3 College of Safety Science and Engineering, Xi'an University of Science and Technology, Xi'an 710054, China
* Correspondence: ycw@stu.xust.edu.cn

**Abstract:** Intelligent mining needs to achieve real-time acquisition of surrounding rock deformation data of roadways and analysis and provide technical support for intelligent mining construction. To solve problems such as significant error, information lag, and low acquisition frequency of surrounding rock monitoring technology at the current stage, a perception method, RSBV of roadway deformation situation, based on binocular vision is proposed, which realizes the dynamic, accurate real-time acquisition of host rocks' relative deformation in a dusky environment. The low illumination image enhancement method is used to preprocess original images, which reduces the impact of low illumination and high dust; the K-medoids algorithm segments the target image, and the SIFT algorithm extracts feature points from the ROI (region of interest). The influence of eliminating background images on the feature point extraction is revealed, and the efficiency of feature extraction is improved; the method of SIFT feature-matching with epipolar constraints is studied, which improves the accuracy and speed of feature points. The roadway deformation characteristics are analyzed, and the reflective target is used as the monitoring point. A roadway deformation acquisition and analysis platform based on binocular vision is built in a dim environment. Zhang's method is selected to calibrate the camera parameters, and stereo rectification is carried out for the target motion image. The adaptability of the RSBV method to different surrounding rock deformation scales is studied and compared with the measurement results of the SGBM algorithm. The results show that the error of the RSBV method is controlled within 1.6%, which is 2.88% lower than the average error of the SGBM algorithm. The average time for processing a group of binocular images is 1.87 s, which is only 20% of the SGBM algorithm. The research result provides a reliable theoretical basis for the real-time and accurate evaluation of the surrounding rock deformation mechanism.

**Keywords:** mining roadway; dusky environment; surrounding rock deformation; binocular stereo vision; image features matching

## 1. Introduction

Delicate perception of the surrounding rock deformation situation of mining roadways is essential to intelligent coal mine construction [1]. In the tunnelling process, the rock mass's original equilibrium state is broken, and the surrounding rock has the potential safety hazard of instability and destruction [2–4]. This often leads to roof collapse, floor heave, sidewall heave, and other accidents, which significantly impact coal mine safety production [5,6]. By collecting the amount of surrounding rock deformation and analyzing its changing trend, the safety condition of the roadway can be evaluated [7,8]. For example, in the article [9], through long-term geotechnical engineering monitoring, it was found that the deformation of roadway roof strata is affected by the support scheme, roof stratification, and rock strength, and the relationship between the support scheme and surrounding rock movement was summarized according to the monitoring results. Currently, the main

roadway deformation monitoring methods ignore the persistence of surrounding rock deformation and the whole movement failure of surrounding rock. There are problems such as significant measurement error, low frequency of mining analysis, and information lag. Therefore, a remote, real-time, continuous monitoring, and accurate measurement method for roadway surrounding rock deformation perception is urgently needed.

Traditional roadway deformation monitoring methods mainly measure the convergence of surrounding rock using manually operated instruments, such as the cross-measurement method and convergence meter monitoring method [10]. Lu et al. [11] used a convergence meter to calculate the convergence of roadway surrounding rock in a roadway deformation monitoring study. Such methods require the manual arrangement of monitoring tools, which have the disadvantages of human interference, low automation, and long time consumption [10,12]. In addition, some high degrees of automation monitoring methods, such as the roof abscission layer instrument measuring method, optical fiber sensing technology, 3D laser measurement technology, etc., are used [10,13–15]. Based on a distributed optical fiber sensing technology, Tang et al. [10] used optical fiber to monitor the real-time deformation and continuous deformation of surrounding rock under the stimulation of a laser pulse. Hou et al. [13] analyzed the application prospect of distributed optical fiber sensing technology in roadway safety monitoring. They believed that distributed optical fiber has long-distance, extensive range and distribution characteristics and is suitable for roadway deformation monitoring. Wang et al. [15] combined 3D laser scanning technology and the convergence instrument measurement method and proposed a 3D visualization method for roadway deformation by taking advantage of the visualization advantages of 3D laser-scanning technology and the advantages of convergence instruments for accurate measurement and easy operation. Kajzar et al. [16,17] used a pulsed three-dimensional laser scanner to monitor the stability of roadways and coal pillars. They obtained the convergence of the roof, floor, two sides of the surrounding rock, and the stability of the coal pillar, which confirmed this method's feasibility in stabilizing the roadway's surrounding rock. Compared with traditional methods, the above measurement methods are more automatic and accurate, but there are still some limitations. Roof separation instruments can measure the internal deformation of surrounding rock, but it is easy for equipment to cause errors caused by, and it is challenging to monitor small deformations accurately. Optical fiber sensing technology requires optical fiber layout, complex wiring, high cost, and contact measurement, with the possibility of missing detection [13]. The 3D laser measurement method has an extensive monitoring range and high accuracy, but it has a large amount of point cloud data, complicated later work, and high equipment cost. In recent years, 3D measurement methods based on stereovision have been widely applied in robot and industrial measurement fields due to their advantages of noncontact, real-time accuracy, and access to scene depth information [18–20]. For example, Zhang et al. [21] built a position and orientation measurement system for the boring machine body based on the three-laser-point target and monocular vision measurement technology, which realized the accurate calculation of the spatial position and orientation of the boring machine. Xu et al. [22] obtained the deformation of the roadway surrounding rock based on a binocular vision algorithm. They established a real-time measurement and warning system for the roadway's surrounding rock. To sum up, it is of great significance for the intelligent and safe mining of coal mines to apply stereo vision technology in roadway deformation monitoring and master the stability state and development trend of surrounding rock through intelligent means. Therefore, this paper focuses on how to apply stereo vision technology to the real-time monitoring of surrounding rock deformation and analyze its measurement effect in practice through roadway deformation measurement experiments.

Therefore, the key research objectives are how to apply stereo vision technology to real-time monitoring of surrounding rock deformation, accurately measure the relative distance of surrounding rock, and analyze its measurement effect in practice using a tunnel deformation measurement experiment. An experimental platform for the roadway section's surrounding rock deformation was designed based on the analysis of roadway deformation

characteristics. A real-time perception method, RSBV of roadway deformation situations, based on binocular stereo vision was proposed. Firstly, the image segmentation method was used to extract the monitoring target image in the complex environment of the roadway, and the disparity was calculated by the feature-matching method with epipolar constraint. Then, the 3D coordinates of the target feature points were calculated according to the binocular vision model, and the relative deformation between the targets was obtained. The rationality and effectiveness of the method were verified by algorithm comparison and error analysis. The research proves the feasibility of applying the binocular stereo vision method in roadway deformation monitoring and provides a specific reference for subsequent roadway deformation measurement methods.

## 2. Principle of Roadway Deformation Measurement

Figure 1 shows common roadway deformation and failure phenomena: Figure 1a, roof collapse; Figure 1b, sidewall bulge; and Figure 1c, floor heave. It can be seen that the failure and deformation of the surrounding rock mainly occur in the roof, two sides, and floor of the roadway [23], and the safety assessment of the roadway is mainly based on the deformation data of these parts.

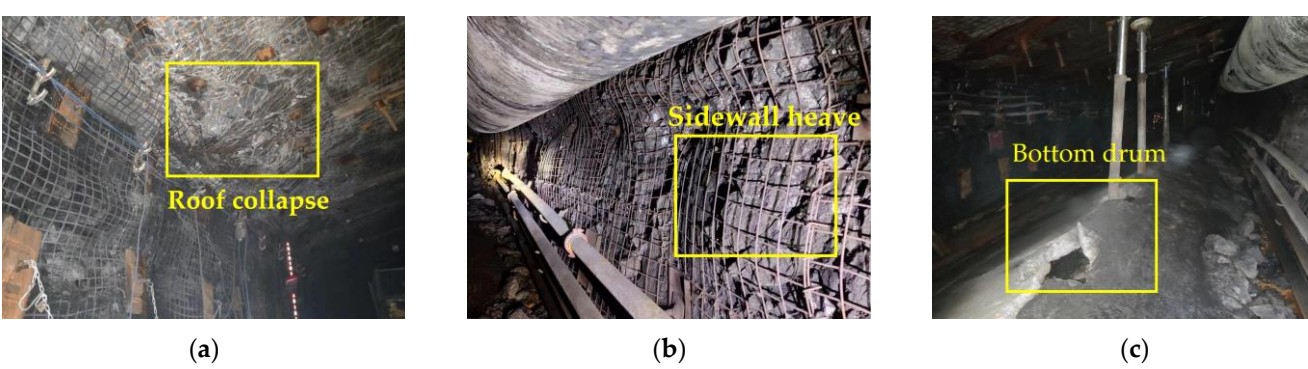

**Figure 1.** Roadway deformation characteristics; (**a**) roof deformation; (**b**) sidewall deformation; (**c**) floor heave deformation.

Based on the above analysis of roadway deformation, the experiment realizes the monitoring of roadway surrounding rock damage by simulating the deformation of the main parts of the roadway. Considering that the four walls of the roadway are covered with anchoring and shotcreting, and the illumination is poor, the image texture directly collected is repeated, and it is not easy to find the appropriate measurement points. Therefore, the reflective target is the monitoring point to provide the measurement basis for binocular vision. In addition, considering that the actual roadway floor needs to work usually, it is challenging to arrange targets for monitoring, so the main research object of the experiment is the host rock part of the roadway [24]. Figure 2 shows five targets on the roadway roof, shoulder angles on both sides, and both sides as measuring points. The red dashed line represents the shape of the roadway and its monitoring targets after deformation. The deformation of the roadway is the convergence of the two sides and the roof and floor. Therefore, the focus can be focused on the target displacement, and the convergence of surrounding rock can be obtained by calculating the relative distance between targets at different times.

According to the binocular stereo vision algorithm, the three-dimensional coordinates of target feature points can be obtained, and the relative distance between targets can be calculated. If the coordinates of the target feature points on the left side of the roadway are $(X_j, Y_j, Z_j)$, and the coordinates of the target feature points on the right side are $(X_j, Y_j, Z_j)$, the relative distance $S$ between the two sides is:

$$S = \sqrt{(X_i - X_j)^2 + (Y_i - Y_j)^2 + (Z_i - Z_j)^2} \tag{1}$$

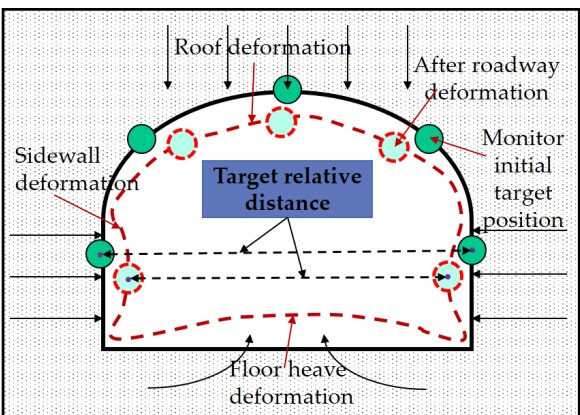

**Figure 2.** Roadway section deformation monitoring diagram.

Based on the construction of the above scene, a real-time perception method RSBV for mining roadway deformation situations based on binocular stereo vision is proposed. The overall technical route is shown in Figure 3, which includes six parts: binocular camera calibration, binocular image acquisition, stereo rectification and preprocessing, ROI extraction, feature point disparity calculation, and 3D coordinate reconstruction. Among them, the disparity calculation is the core step, and the SIFT feature matching method is used to calculate the disparity. Since SIFT feature point detection is slow, and the ROI area of the experimental study is only the target image, too many unnecessary background feature points increase the false matching rate. Therefore, considering reducing the search range of SIFT feature points and improving the algorithm's speed, the K-medoids algorithm was used to segment the target image region (ROI) and extract the feature points. At the same time, to locate the target roughly, the Canny algorithm was used to extract edge pixels of the target image to fit, and the center coordinates were calculated according to the fitting equation. Then, the feature matching method fused with epipolar constraint matched the corresponding feature points in the left and right images. The PROSAC (progressive sample consensus) algorithm was used to reduce the false matching points and improve the speed and accuracy of disparity calculation. After calculating the disparity value of the target feature points, combined with the binocular vision model, the 3D coordinates of the target could be calculated. Then, the deformation monitoring of the roadway's surrounding rock could be realized by calculating the relative distance between the feature points.

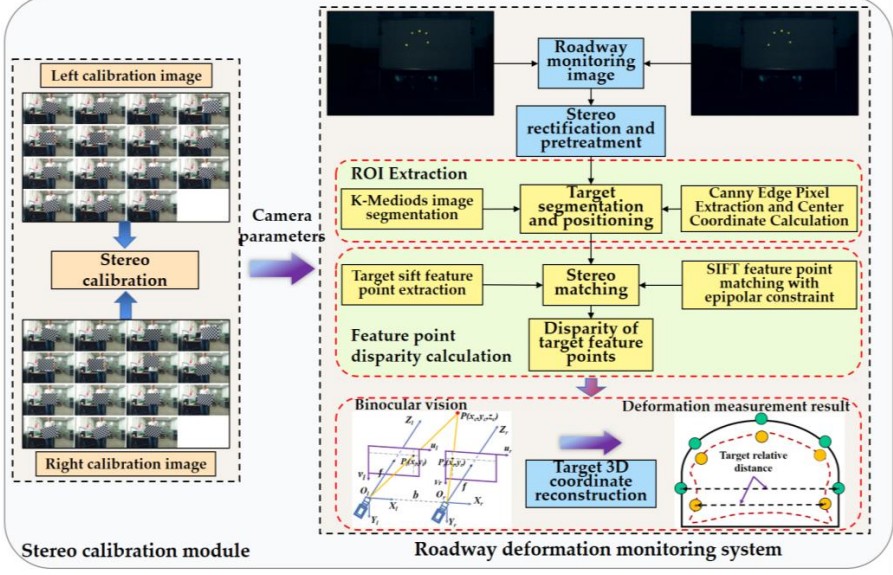

**Figure 3.** Technical route of roadway deformation monitoring system.

### 3. ROI Segmentation and Center Point Coordinate Calculation of Roadway Monitoring Target Image

In the actual working environment of the mining roadway, due to the lack of a light source and high dust concentration, the collected images have low brightness and blur problems. Therefore, first, the collected monitoring image was preprocessed to enhance the low-illumination image and reduce the impact of dust. In addition, considering that the actual roadway monitoring environment, the influence of light, material, and other factors lead to the uneven distribution of the collected image brightness, forming too many useless feature points, which affects the matching effect of feature points. According to the actual environment of the coal mine and combined with the experimental design, the ROIs were segmented using the K-medoids algorithm, and the segmentation results were optimized using the connected region detection method. Then, the Canny edge detection algorithm and ellipse fitting equation were used to locate the target roughly, determine the location of the target center, and reduce the search range of SIFT features. Furthermore, the coordinates of the center point obtained by fitting could be approximated as the computed coordinates of the disparity in the absence of feature points.

#### 3.1. Binocular Image Preprocessing

In the actual mining roadway, a large amount of dust is produced due to the working face mining. In addition, due to the weak light, the images collected were low illumination images, which needed to be enhanced. In the preprocessing step, the method proposed by Xuan [25] was adopted to improve the brightness of low-illuminance images collected. Researchers found that the low-illumination image has some commonalities with the foggy image in the distribution of pixel values after inversion. Using this feature, the low-illuminance images after inversion were processed by using the defogging algorithm. Finally, the image after brightness enhancement could be obtained by reinverting the processing results. In addition, since the underground humid air and high dust concentration environment had a similar effect on the image to the effect of dense fog, the defogging algorithm could be used again to remove the influence of dust after enhancing the low illumination image.

For the defogging algorithm, the dark channel prior defogging algorithm was used. The algorithm is based on the atmospheric scattering model, and the enhancement formula of the defogging image is shown in Equation (2).

$$J(x) = \frac{I(x) - A}{t(x)} + A \tag{2}$$

where $I(x)$ refers to the foggy map, $J(x)$ refers to defogging map, $A$ refers to the atmospheric light value, $t(x) = e^{-\beta d(x)}$, $\beta$ represents the scattering coefficient of the atmosphere, and $d(x)$ represents the depth of field of the image.

It can be seen from the formula that the most important is the estimation of atmospheric light values $A$ and $t(x)$. According to the dark channel prior knowledge, the values of $t(x)$ and $A$ can be estimated from the foggy image $I(x)$, and then the dehazed image $J(x)$ can be obtained.

#### 3.2. ROI Segmentation Method Based on K-Medoids Algorithm

The K-medoids algorithm is an unsupervised clustering algorithm that can divide sample data into several categories according to the distance relationship between samples without sample labels [26]. Equation (3) is the loss function of this algorithm, which can be defined as the sum of squares of distance errors between each sample point and each clustering center. The condition for clustering to end is to minimize the loss function [27]. The ROI of the roadway section monitoring image was segmented based on the K-medoids method. In this experiment, ROI refers to the image containing only the monitoring target, but not the background part. The feature point detection step only extracts SIFT feature points for target images, so this part of the images is the region of interest (ROI). As shown

in Figure 4, the target image region marked by the blue box is the ROI to be extracted. In the subsequent processing, background images were removed, and only ROI images were retained.

$$E = \sum_{i=1}^{K} \sum_{x \in C_i} \left\| x - u_i \right\|_2^2 \tag{3}$$

where $x$ is the sample point; $u_i$ is the center point of the $i$-th class, calculated by the median; $C_i$ represents the $i$-th class; $K$ represents the number of cluster centers.

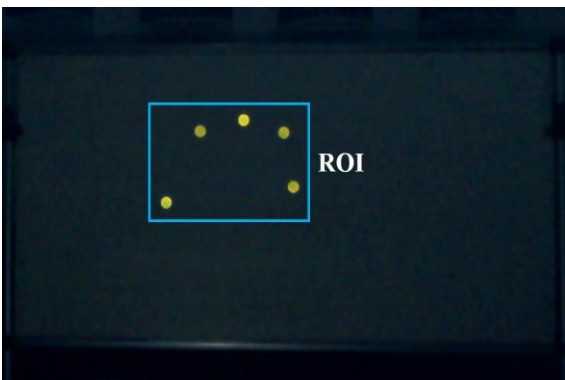

**Figure 4.** Monitor the region of interest of the image.

K-medoids is an improvement of the K-means algorithm, which uses the median of sample data instead of the mean as the clustering center. When there are noises and isolated points in the image, K-medoids are more robust and more suitable for image segmentation. In addition, for the problem that the algorithm randomly selects the initial clustering center, which easily causes local optimization, the optimization idea of K-means++ is introduced. When initializing the clustering center, the distance between the clustering centers is calculated to make the distance between the selected clustering centers as far as possible. In addition, due to the complexity of the background image, it is difficult for the K-medoids algorithm to cluster all pixels according to the ideal result. The connected region detection method was used to eliminate the small area and noise points. ROI extraction steps using the improved K-medoids algorithm are as follows:

1.  A Gaussian filter was used to smooth the binocular image to reduce the impact of noise on image segmentation;
2.  The improved K-medoids algorithm was used to extract ROI from left and right images, and the image was divided into target and background images;
3.  We detected the connected region of the segmented image to remove the connected domain and noise points with the small area;
4.  We restored the original pixel of the target image.

*3.3. Target Image Edge Detection and Center-Point Coordinate Calculation*

The circular target was an ellipse after imaging through the lens, so the edge pixel of the target could be extracted and positioned first, and then the equation of the ellipse could be fitted according to the edge pixel. Then, the center coordinate of the ellipse could be further solved, that is, the position of the center point of the target image.

Using the Canny algorithm to extract target image edge pixels: First, remove the image noise, then find the intensity gradient of the image, use the non-maximum suppression method to eliminate the false edge according to the gradient intensity of the pixel; then, use the double threshold detection to find the potential edge, and finally, use the lag technology to track the edge to get a single pixel edge image [28]. Then, according to the ellipse's general equation (Equation (4)), use the least square method to fit the edge pixels [29],

calculate five unknown parameters in the equation, and then calculate the fitting function of the target edge.

$$Ax^2 + Bxy + Cy^2 + Dx + Ey + 1 = 0 \tag{4}$$

In the equation, $(x, y)$ is the coordinate of the target image edge pixel; $A$, $B$, $C$, $D$, and $E$ are coefficients of any elliptic equation.

Finally, the coordinates of the center point of the target image $(X_{center}, Y_{center})$ were calculated according to Equation (5) of the center of elliptic geometry.

$$\begin{cases} X_{center} = \frac{BE-2CD}{4AC-B^2} \\ Y_{center} = \frac{BD-2AE}{4AC-B^2} \end{cases} \tag{5}$$

## 4. Binocular Stereo Vision Measurement Algorithm

### 4.1. Three-Dimensional Coordinate Calculation Principle of the Roadway Monitoring Target

Similar to human eyes acquiring scene depth information, the binocular vision method is based on the principle of disparity. Two cameras on the left and right were used to shoot the same object. There was a position gap between the pixels of the left and right images, which is a disparity. By calculating the disparity between corresponding points in the image, three-dimensional information about the object could be obtained using the triangulation principle [30]. The model of using stereo vision to calculate the three-dimensional coordinates of target feature points is shown in Figure 5.

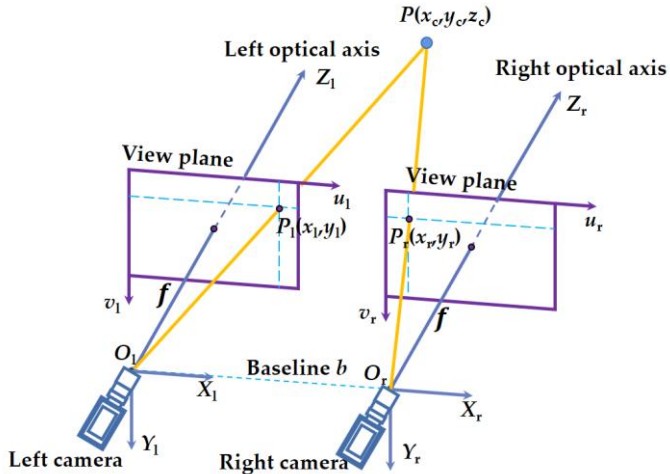

**Figure 5.** Three-dimensional coordinate calculation principle of monitoring target feature points.

The figure was a parallel binocular vision model under ideal conditions; that is, the optical parameters of the left and right cameras were consistent, the left and right optical axes were parallel, and the left and right view planes were coplanar. The roadway surrounding rock's monitoring point $P(x_c, y_c, z_c)$ is projected on the left and right imaging planes after passing through the optical centers $O_l$ and $O_r$ of the left and right cameras. The projection points are $P_l(x_l, y_l)$ and $P_r(x_r, y_r)$. Then, the *disparity* between the projection points $P_l$ and $P_r$ is the distance between the two points on the *x*-axis coordinate, as shown in Equation (6).

$$disparity = x_l - x_r \tag{6}$$

where the unit of *disparity* is pixels.

From the disparity value combined with the similar triangle theorem, the three-dimensional coordinates of point $P$ in the coordinate system with the optical center of the left camera as the origin can be calculated as:

$$\begin{cases} x_c = \frac{bx_l}{disparity} \\ y_c = \frac{by_l}{disparity} \\ z_c = \frac{bf}{disparity} \end{cases} \tag{7}$$

where $f$ is the focal length of the camera, representing the distance between the camera's optical center and the projection plane; $b$ is the baseline, representing the distance between the left and right cardiac $O_l$ and $O_r$.

### 4.2. Camera Calibration Principle

In order to determine the transformation relationship between 2D image pixel coordinates and world 3D coordinates, it is necessary to obtain the parameters related to the transformation through camera calibration to complete the reconstruction of the 3D scene. Figure 6 shows the imaging process of a commonly used pinhole camera model.

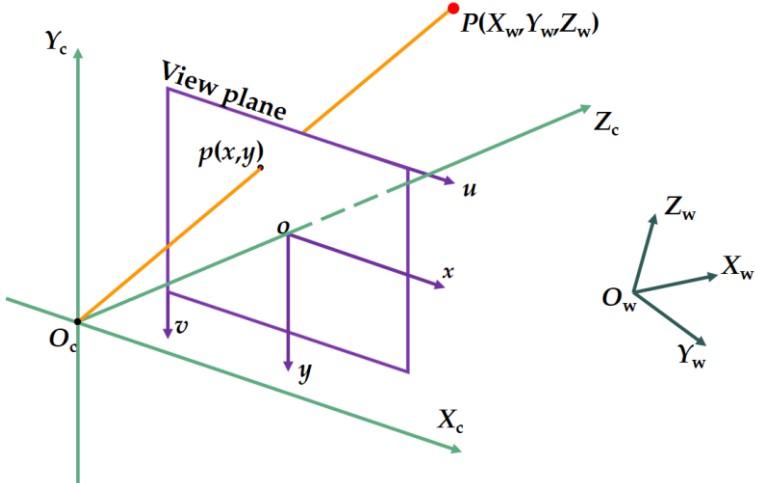

**Figure 6.** Pinhole camera imaging model.

The imaging process is as follows: a point $P(X_w, Y_w, Z_w)$ in the world coordinate system $O_w$-$X_w Y_w Z_w$ passes through the camera optical center $O_c$ and is imaged on the projection plane to form a projection point $p(x, y)$ whose coordinates in the camera coordinate system $O_c$-$X_c Y_c Z_c$ are $(x_c, y_c, z_c)$. Generally, the projection transformation process from space point $P$ to pixel point $p$ is represented by Equation (8) [31].

$$s \begin{bmatrix} x \\ y \\ 1 \end{bmatrix} = \begin{bmatrix} f_x & 0 & u_0 & 0 \\ 0 & f_y & v_0 & 0 \\ 0 & 0 & 1 & 0 \end{bmatrix} \begin{bmatrix} R & T \\ \vec{0} & 1 \end{bmatrix} = K \begin{bmatrix} R & T \\ \vec{0} & 1 \end{bmatrix} \begin{bmatrix} X_w \\ Y_w \\ Z_w \\ 1 \end{bmatrix} \tag{8}$$

where $s$ is any scaling factor; $K$ is the camera internal parameter matrix; $R$ and $T$ are the rotation and translation matrix of the right camera with respect to the left camera transformation, called the external parameter matrix of the camera; $f_x = f/dx$, $f_y = f/dy$ are the normalized focal lengths on the $x$ and $y$ axes, respectively, in the pixel; $u_0$ and $v_0$ represent the intersection of the optical axis and the projection plane, which is called the "principal point".

The distortion factor should also be considered in practical application. Due to the manufacturing process and installation accuracy of the camera lens, the image taken by the

camera lens has a certain degree of distortion, and the image containing distortion reduces the matching accuracy in stereo matching. Figure 7a is a standard image without distortion, while Figure 7b,c are common pincushion distortion and barrel distortion.

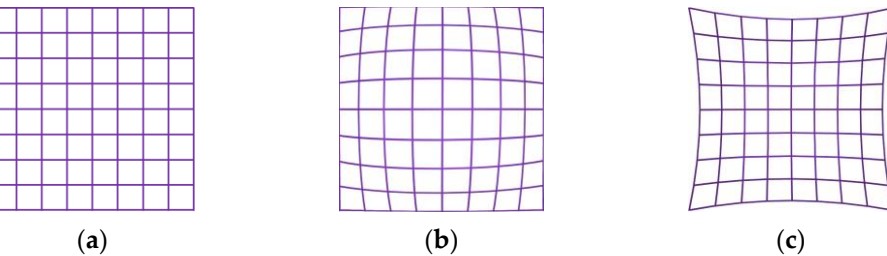

**Figure 7.** Images of different distortion types; (**a**) undistorted image; (**b**) barrel distortion; (**c**) pincushion distortion.

The pinhole imaging model mainly considers radial distortion and tangential distortion. The lens shape causes radial distortion, and the error causes tangential distortion in the lens installation process. The radial distortion model in Equation (9) and the tangential distortion model in Equation (10) can be established [32].

$$\begin{cases} x_{distorted} = x(1 + k_1 r^2 + k_2 r^4 + k_3 r^6) \\ y_{distorted} = y(1 + k_1 r^2 + k_2 r^4 + k_3 r^6) \end{cases} \tag{9}$$

$$\begin{cases} x_{distorted} = x + [2p_1 xy + p_2(r^2 + 2x^2)] \\ y_{distorted} = y + [p_1(r^2 + 2y^2) + 2p_2 xy] \end{cases} \tag{10}$$

where $(x, y)$ is the original position of the distorted pixel, $(x_{distorted}, y_{distorted})$ is the corrected position; $r$ is used for Taylor series expansion; $k_1$, $k_2$, $k_3$ are radial distortion coefficients, $p_1$, $p_2$ are tangential distortion coefficients.

The five commonly used distortion coefficients $k_1$, $k_2$, $k_3$, $p_1$, and $p_2$ are the distortion parameters to be determined in camera calibration experiments.

### 4.3. Binocular Stereo Rectification

Since the image captured by the camera was distorted and the imaging plane was not coplanar and in line alignment, it was necessary to carry out distortion correction and stereo rectification on the binocular image to make it reach the ideal binocular stereo vision model. The binocular stereo rectification process is shown in Figure 8.

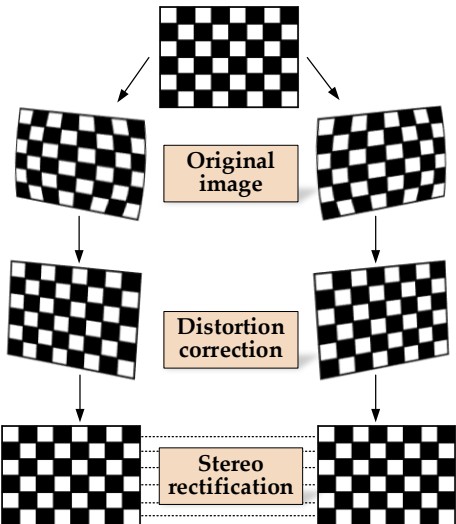

**Figure 8.** Binocular stereo rectification flow chart.

The image distortion correction methods used are as follows: perspective transformation matrix was calculated from radial distortion coefficient and tangential distortion coefficient, and a distortion-free image was obtained by linear interpolation and remapping.

Stereo rectification includes two steps: binocular image coplanar and line alignment. The left of Figure 9 shows the geometric constraints in the actual imaging. The intersection lines $p_l e_l$ and $p_r e_r$ of the polar plane $O_L O_R P$ and the image plane are called polar lines [33]. According to the principle of polar geometry, the projection point of point $P$ on the left imaging plane is $p_l$, so its projection point on the right imaging plane must be on the corresponding polar line. According to this principle, feature matching only needs to be searched on the polar $p_r e_r$, but the efficiency of this method is relatively low. Through stereo rectification, the polar lines of the two images are located in the same plane and aligned horizontally. The 2D search is reduced to 1D, which can improve the matching efficiency. The effect after rectification is shown in the right model in Figure 9.

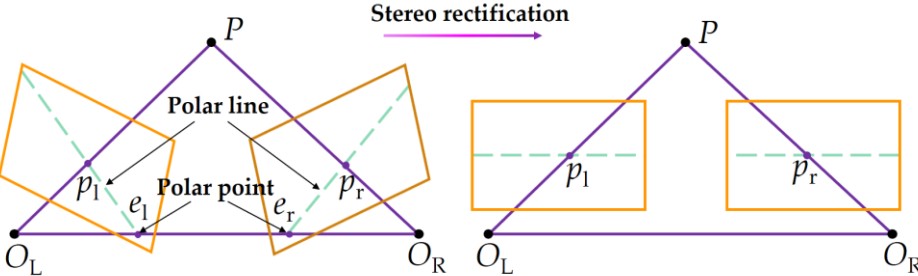

**Figure 9.** Stereo rectification diagram.

The stereo rectification uses the bought algorithm. By calculating the reprojection matrix of each camera, the binocular images are reprojected on the same plane, making the two images coplanar. The reprojection matrix is shown in Equation (11).

$$Q = \begin{bmatrix} 1 & 0 & 0 & -c_x \\ 0 & 1 & 0 & -c_y \\ 0 & 0 & 0 & f \\ 0 & 0 & \frac{-1}{T_x} & \frac{c_x - c'_x}{T_x} \end{bmatrix} \tag{11}$$

where $c_x$ and $c_y$ are the coordinates of the principal point of the left camera; $c'_x$ is the principal point of the right camera on the *x*-axis; $T_x$ is the translation vector of the right camera with respect to the left camera.

### 4.4. Stereo Matching

Stereo matching calculates the disparity by finding the pixels with the same characteristics in the left and right images, that is, to find the projection points of the object points on the left and right projection planes, which is the core step of binocular vision. SIFT feature point matching is used to calculate the disparity, and the results are compared with SGBM semiglobal stereo matching algorithm. SIFT feature point matching is a sparse matching method that calculates the disparity by matching the detected SIFT feature points. The SGBM algorithm can perform dense matching and consumes less time than global matching. It can obtain the binocular image's disparity map, which stores each pixel's disparity value. The final research target only needed to calculate the disparity value of the monitoring target feature points without dense matching, so the SIFT algorithm was selected to complete the feature point matching.

4.4.1. SIFT Algorithm

1. Feature point detection

SIFT is a kind of image local feature descriptor independent of image rotation and scaling and has a high tolerance to illumination change. It can accurately detect the SIFT feature

points of the target under different illumination and positions. Compared with the general corner points and feature points, the robustness is stronger in complex cases. Moreover, the SIFT algorithm can accurately extract subpixel coordinates for disparity calculation.

SIFT feature detection mainly includes four steps [34]:

- Scale space extremum detection;
- Feature point localization;
- Determine the direction of each feature point;
- Feature point descriptor generation.

SIFT algorithm constructs a scale space for two-dimensional images, uses the Gaussian difference method to detect feature points, and searches for extreme points in different scale-spaces to obtain scale invariance; in order to achieve rotation invariance, the direction histogram is introduced to describe the main direction of feature points to detect feature points in different directions; by selecting the surrounding 4 × 4 fields for each feature point as the sampling window, and calculating the gradient histogram of eight directions for each window, the 128-dimensional feature descriptor is generated. Based on this, SIFT has good robustness for image illumination. Figure 10 shows the SIFT feature points and their descriptors on the roadway target image. By selecting 4 × 4 size windows near the feature points, each window uses gradient histograms of eight directions to describe the local gradient and finally obtain SIFT feature descriptors of 128 directional histograms.

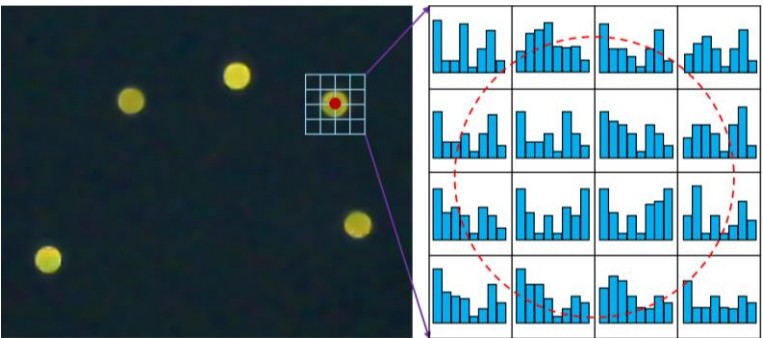

**Figure 10.** SIFT features descriptor.

2. SIFT Feature Matching with Epipolar Constraint

The traditional SIFT feature point matching method performs feature matching by traversing and searching the point to be matched closest to the feature point in the right image and the left image. This method needs to traverse the feature points in the whole image to be matched during matching, which has a large search range, a long matching time, and is easy to cause false matching. Based on the results of stereo rectification, this study improves the traditional matching method and proposes a SIFT feature point matching method with epipolar constraint.

The epipolar constraint means that for a feature point in one image, the point to be matched in another must be located on the corresponding polar line. When matching SIFT feature points, for the feature points in the left image, it is only necessary to search for the points to be matched on the corresponding polar line in the right image and the epipolar constraint Equation (12) between the feature points $p$ and $p'$ can be established.

$$p'^{T}Fp = 0 \tag{12}$$

where $F$ is the basic matrix of the camera.

Stereo rectification realizes the coplanar and line alignment of binocular images by rectification the polar lines. In feature matching, adding epipolar constraints can narrow the search range of feature points, and the corresponding matching points can be found in the same line. The two-dimensional search between two images is transformed into a

one-dimensional search along the same line, which significantly improves the accuracy and speed of SIFT feature point matching.

Feature point matching uses a FLANN matcher, which provides a variety of methods to find the nearest neighbor points in high-dimensional space and can be used for matching feature points with close distances [35]. Finally, the PROSAC algorithm is used to remove mismatched points. PROSAC algorithm is an improvement of the RANSAC (random sample consensus) algorithm. Compared with RANSAC, which evenly extracts samples from the whole set of matching points, the PROSAC algorithm first sorts the Hamming distance between the matching points and samples from the matching points close to each other, which can reduce the amount of calculation and improve the algorithm speed.

### 4.4.2. SGBM Algorithm

The SGBM algorithm is implemented in OpenCV using Hirschmuller's SGM algorithm. SGBM is an improvement of the SGM algorithm, a semiglobal stereo matching method. The algorithm is divided into four steps: preprocessing, cost calculation, dynamic programming, and post-processing. By establishing a sliding window and using the dynamic programming method, the cost of pixels in the window is calculated one by one to get the optimal matching results. This method has high precision of disparity calculation and good robustness and is often used for the 3D reconstruction of objects.

## 5. Experimental Results and Analysis

Due to the slow deformation of roadway surrounding rock in the actual working environment, large deformation does not occur in the short term. Therefore, this experiment adopts the artificial simulation of roadway deformation to complete the measurement experiment. Figure 11 shows the roadway deformation simulation experiment platform and test scene, mainly composed of image acquisition equipment, the roadway simulation section, and a binocular vision measurement system.

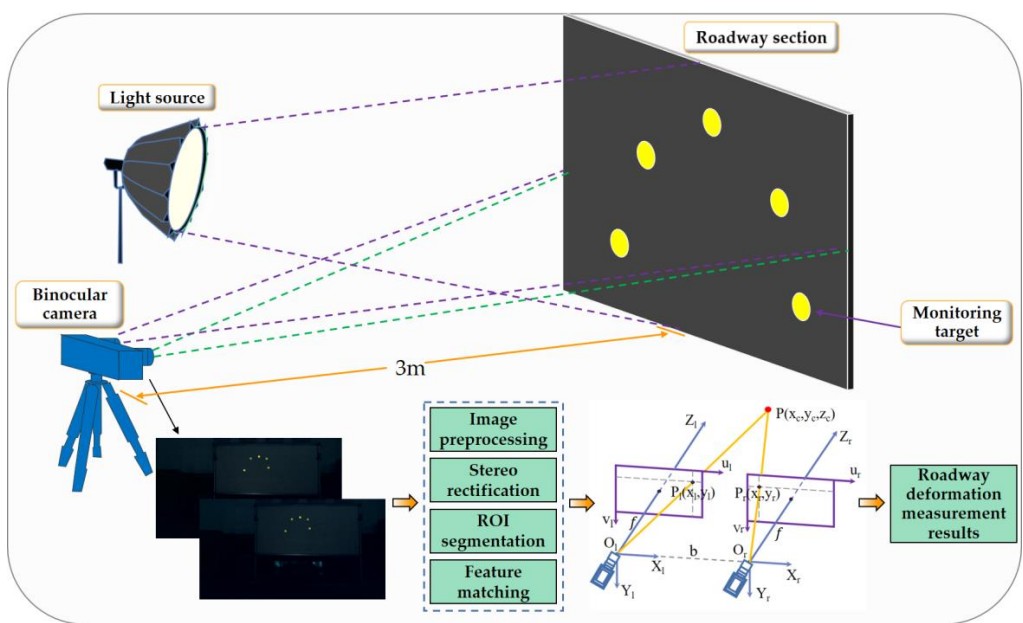

**Figure 11.** Experimental Platform of Roadway Deformation Monitoring System.

The primary purpose of this experiment is to test the measurement effect of the binocular vision algorithm in actual roadway deformation monitoring. The reflective target was arranged on the roadway section to assist in the deformation monitoring of the surrounding rock of the roadway. The actual diameter of the target was 49 mm. The deformation displacement of the roadway host rocks was simulated by manually moving

the target; the experiment simulated the dark environment of the roadway and selected the LED lamp commonly used in the coal mine as the auxiliary light source of the image acquisition equipment. The error of the binocular camera used in the experiment was less than 5‰ when the distance was less than 3 m. In order to reduce the accumulation of errors, the binocular camera and the control computer took pictures at a distance of 3 m in front of the roadway section to collect binocular images. A binocular vision measurement system processed the acquired images to obtain the relative distance between the target features. In this experiment, a Pixel XYZ®DUAL-100M-60 binocular camera was adopted, with a baseline distance of 60 mm and a maximum resolution of 1280 pixel × 720 pixel, which is close to the resolution of the actual mine camera.

### 5.1. Binocular Camera Calibration Experiment

Zhang's calibration method was used for camera calibration [36]. This method only needs a plane chessboard, and the camera's internal and external parameters can be calculated according to the relationship between the points on the chessboard and the imaging corners. Because of its high calibration accuracy, simple operation, and strong robustness, it is widely used in the practical working environment. The binocular camera and calibration images used in the experiment are shown in Figure 12.

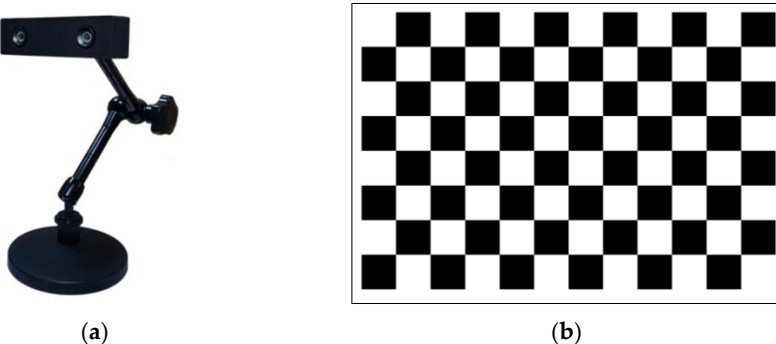

(**a**) (**b**)

**Figure 12.** Camera calibration experimental equipment; (**a**) binocular camera; (**b**) checkerboard image for calibration.

According to Zhang's calibration method, 29 sets of binocular images were taken for calibration experiments. Firstly, corner points in binocular calibration images were extracted using the MATLAB calibration toolbox, then images with large errors were deleted according to reprojection errors. The remaining 15 groups of images were used for final calibration. The reprojection errors are shown in Figure 13. The image displays the error between the detected corner point and the actual position of each image in the form of a color-coded cross shape. The reprojection error of the left and right groups of images is small and convergent, which meets the requirements of experimental accuracy.

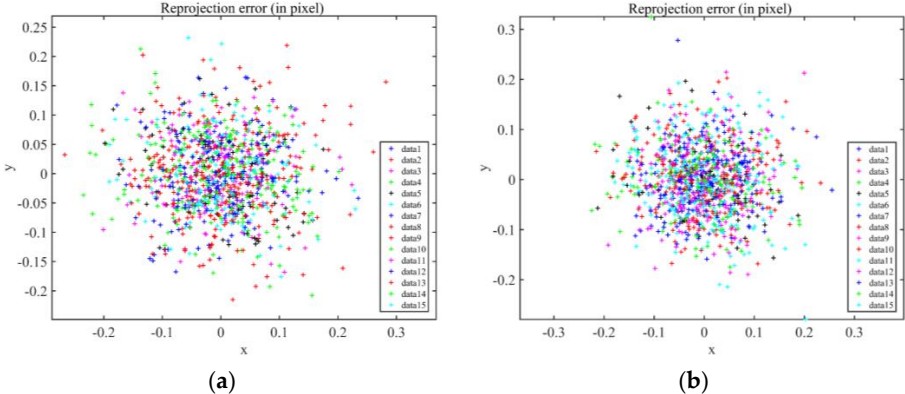

(**a**) (**b**)

**Figure 13.** Reprojection error; (**a**) left reprojection error; (**b**) right reprojection.

After completing the calibration of the binocular camera, the internal parameter matrix, distortion parameters, and external parameters of the left and right cameras were obtained, as shown in Table 1.

**Table 1.** Binocular camera calibration results.

| Camera Parameters | Left Camera | Right Camera |
|---|---|---|
| Intrinsic matrix | $\begin{bmatrix} 891.379 & 0 & 619.270 \\ 0 & 891.052 & 313.978 \\ 0 & 0 & 1 \end{bmatrix}$ | $\begin{bmatrix} 894.317 & 0 & 620.552 \\ 0 & 894.820 & 316.604 \\ 0 & 0 & 1 \end{bmatrix}$ |
| Radial distortion | (0.08173, −0.15417, 0) | (0.08913, −0.27026, 0) |
| Tangential distortion | (0.00123, −0.00406) | (0.00147, −0.00258) |
| Translation vector | (−60.17687, 0.19712, 1.91753) | |
| Rotational vector | (0.00413, 0.00032, −0.00074) | |

*5.2. ROI Segmentation and Location Results*

After the camera calibration was completed, the acquired binocular images were subjected to stereo rectification and preprocessing. The original left image is shown in Figure 14a. On the experimental plane, five monitoring targets were arranged along the roadway contour at the deformable part. The bought algorithm was used for stereo rectification, and the low illumination image enhancement method based on the dark channel defogging algorithm was used to improve the brightness of the binocular image. The processed image is shown in Figure 14b. The edge part of the image was removed after reprojection due to distortion, and the image's brightness was significantly improved compared with the original image. After stereo rectification and preprocessing, the K-medoids algorithm was used to segment ROI in the image, and the segmentation results are shown in Figure 15a. However, the K-medoids algorithm is not ideal for ROI segmentation results, and there were still some noise points and small connected areas. Then the connected region detection method was adopted to remove the small area and restore the original target pixels, and the results are shown in Figure 15b. After optimization, only the original target image was retained.

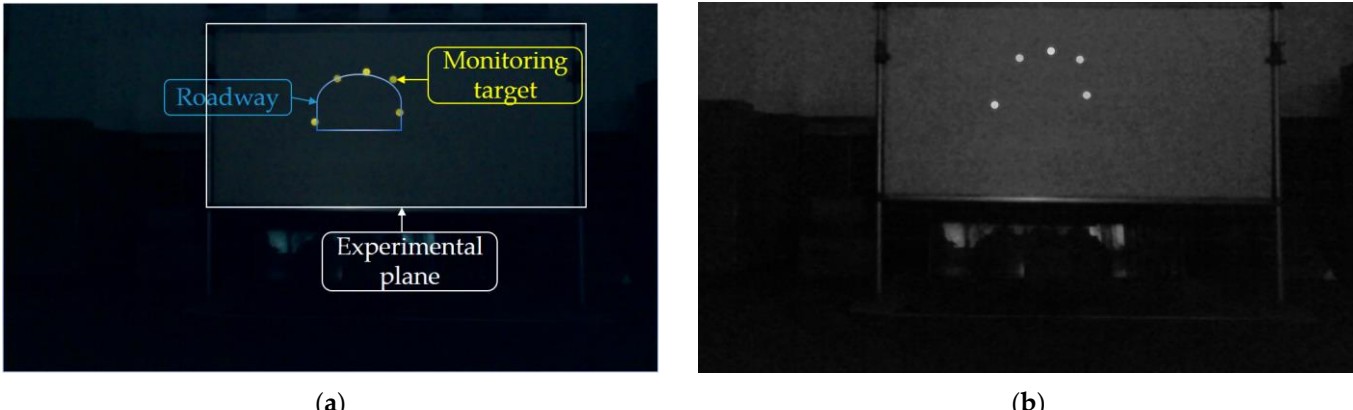

(**a**)  (**b**)

**Figure 14.** Stereo rectification and preprocessing; (**a**) original left image; (**b**) left image after rectification and preprocessing.

After the target region was segmented, the Canny edge detection algorithm and ellipse fitting algorithm are used to locate the target image. Table 2 shows the target center coordinates calculated using the ellipse fitting equation and manually located. Since the SIFT feature point in the experiment was located in the center of the target image, its coordinates were also approximately target center coordinates, so the fitted coordinates could replace the feature point coordinates to complete the calculation when the SIFT

feature point was missing. It can be seen from the table that the coordinates fitted by the elliptic equation are basically consistent with the actual coordinates, which proves that the method can accurately locate the target position. Figure 16 shows the "+" symbol drawn in the Canny edge detection image according to the fitting positioning results. The target center coordinates obtained by the fitting equation are basically in the center position of the ellipse.

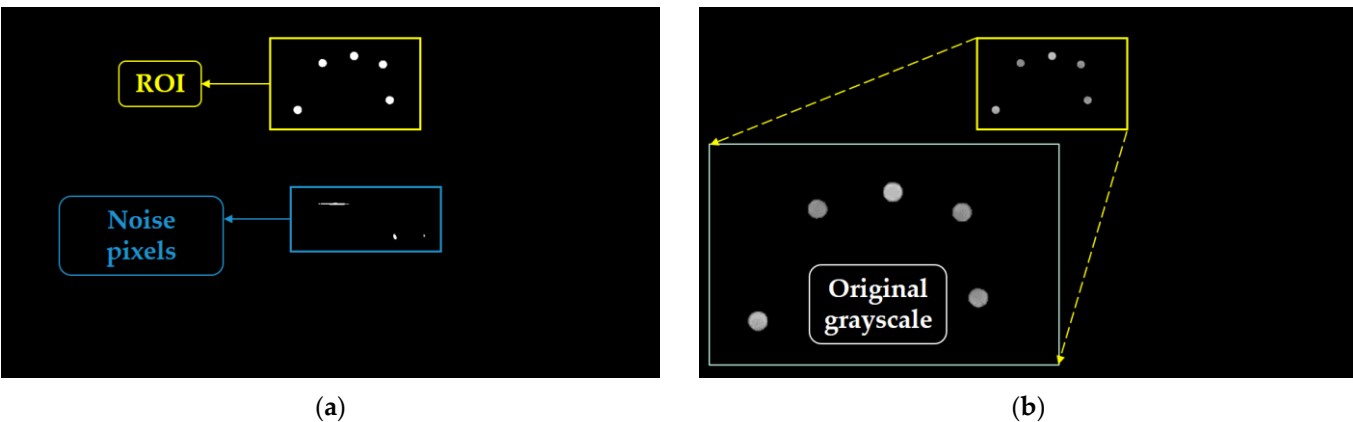

(**a**)            (**b**)

**Figure 15.** Left image ROI segmentation results; (**a**) after K-medoids segmentation; (**b**) after removing noise pixels.

**Table 2.** Target center point coordinates.

| Calculation Method | Coordinate | Target 1 | Target 2 | Target 3 | Target 4 | Target 5 |
|---|---|---|---|---|---|---|
| Fitting result | u | 576.07 | 624.27 | 685.51 | 741.72 | 754.93 |
| | v | 199.06 | 108.52 | 94.81 | 111.07 | 180.94 |
| Manual positioning | u | 576 | 624 | 685 | 741 | 754 |
| | v | 200 | 109 | 95 | 112 | 181 |

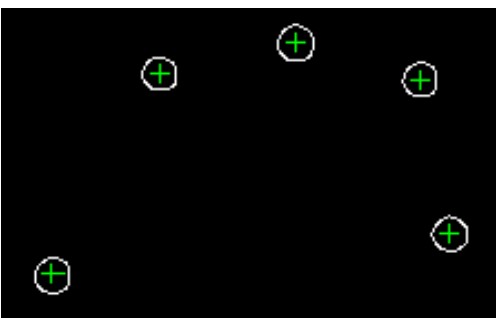

**Figure 16.** Target center point fitting positioning results.

Based on the above ROI segmentation and target center point fitting results, the method used in this experiment can accurately segment and locate target images, providing good experimental conditions for feature detection and disparity calculation.

*5.3. SIFT Feature Point Extraction and Matching Results Analysis*

Figure 17 shows the results of extracting feature points from the original image using SIFT algorithm and extracting feature points only from ROI. The feature points detected in the original image were many and stacked, which makes it mismatching easy. On the other hand, the number of feature points extracted from ROI was reduced, the size similar to the

target image, and all of them were located at the target center, which is conducive to the accurate matching of feature points. In terms of detection speed, extracting feature points from ROI was 80 ms faster than extracting feature points from the original image.

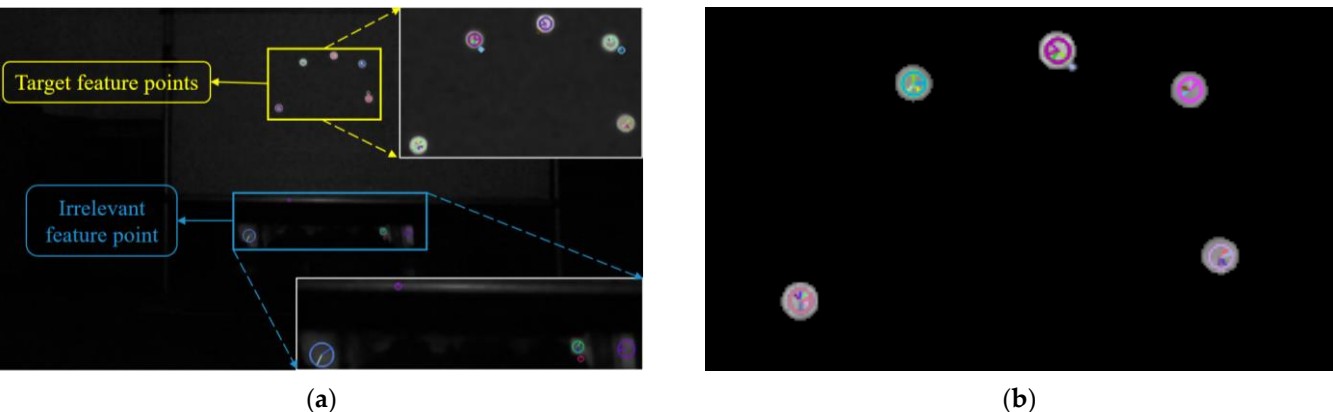

<div align="center">(<b>a</b>)        (<b>b</b>)</div>

**Figure 17.** Feature point extraction and comparison; (**a**) SIFT algorithm extracts feature points from the original image; (**b**) SIFT algorithm extracts feature points from ROI.

After extracting SIFT feature points from ROI, the feature points in the left and right images were matched by the feature matching method fused with epipolar constraint. Figure 18 shows the results of using the traditional SIFT feature-matching method and the feature-matching algorithm proposed in this research. It can be seen from the figure that the traditional feature-matching algorithm calculates the distance of feature points on different epipolar lines to obtain the matching results, and there are many mismatching points. However, in the results obtained using the feature matching algorithm with fusion epipolar constraints in this experiment, the feature points were only searched and matched in the same line, and the matched feature points were all located in the target image, and there were no mismatching points. The matching accuracy is much higher than that of the traditional algorithm. In terms of matching speed, the method used in this experiment was 770 ms faster than the traditional SIFT matching method.

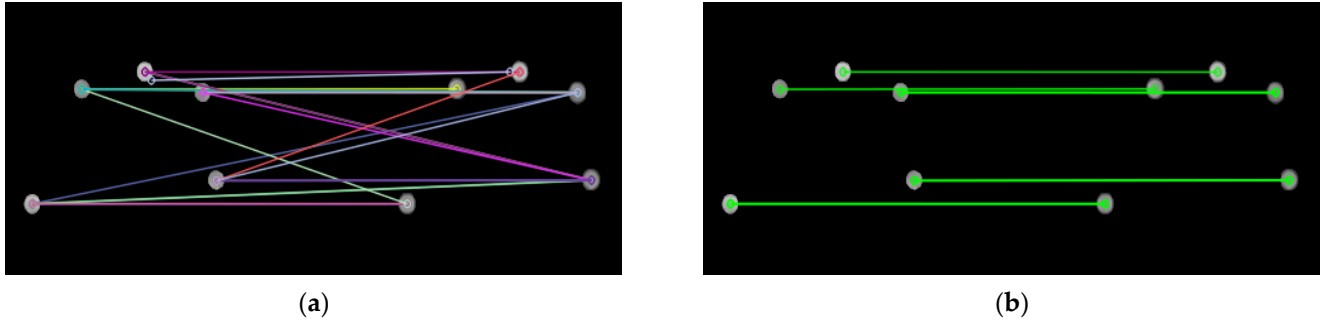

<div align="center">(<b>a</b>)        (<b>b</b>)</div>

**Figure 18.** Feature point matching comparison: (**a**) traditional feature matching algorithm; (**b**) feature matching algorithm in this research.

*5.4. Measurement Results and Analysis of Roadway Deformation*

After feature-matching, the corresponding disparity was calculated from the sub-pixel coordinates between the left and right matching feature points. Then, the three-dimensional coordinates of the target feature points were calculated from the binocular vision model. Finally, the relative distance between target feature points was obtained according to the distance formula. The accuracy of the RSBV method was verified by comparing it with the relative distance between targets measured manually. The experiment verified the advantages of the RSBV method by comparing it with the SGBM algorithm. Figure 19 shows the disparity map processed by the SGBM algorithm. The disparity of the center

point of the target image can be obtained by combining it with the center coordinates of the ellipse calculated by fitting. In order to test the adaptability of the RSBV method under different deformation scales of the roadway, five targets were moved 100 mm and 150 mm in random directions when the distance between the fixed camera and the roadway section was 3 m, as shown in Figure 20. Binocular images of three target groups under different deformation conditions were collected, and roadway deformation measurement experiments were carried out with these image data.

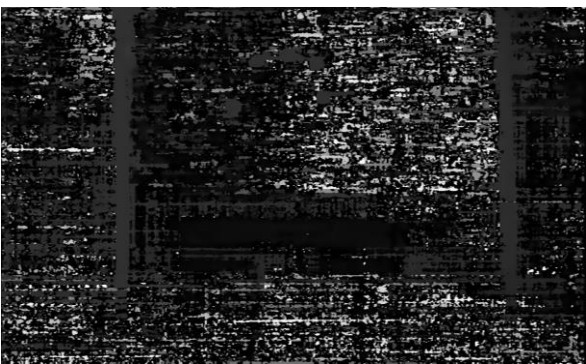

**Figure 19.** Disparity map of SGBM algorithm.

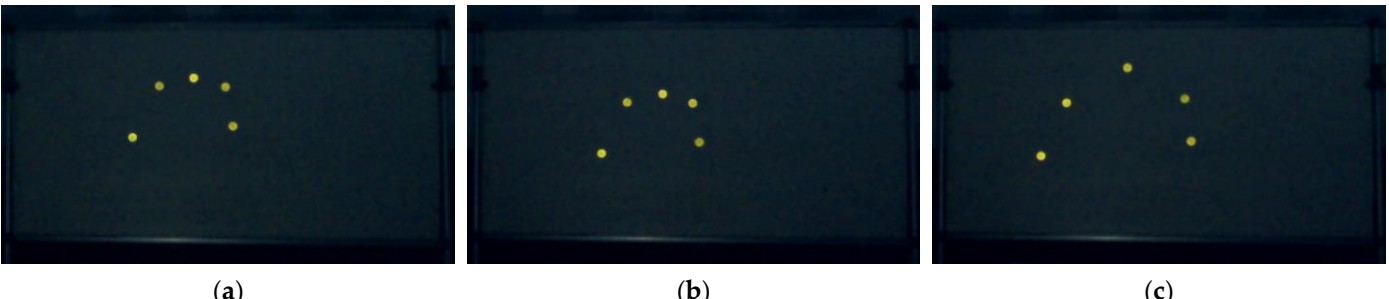

<table>
<tr><td>(**a**)</td><td>(**b**)</td><td>(**c**)</td></tr>
</table>

**Figure 20.** Three groups of target images collected in the experiment: (**a**) initial position of target; (**b**) target moving 100 mm; (**c**) target moving 150 mm.

Three sets of disparity in Table 3 were obtained by stereo-matching three sets of images using SGBM and RSBV methods, respectively. It can be seen from the table that the disparity values calculated by the two algorithms are stable around 20, and the disparity values change a little before and after moving. This is because the depth distance $Z$ is inversely proportional to the disparity, so when the depth distance is basically unchanged, the disparity value changes in a small range. This is consistent with the experiment. The roadway section is basically at the same depth, and the disparity changes little even after moving the target.

**Table 3.** Disparity calculation results.

| Moving Distance | Algorithm | Target 1 | Target 2 | Target 3 | Target 4 | Target 5 |
|---|---|---|---|---|---|---|
| 0 mm | RSBV | 20.06 | 20.13 | 20.10 | 20.06 | 20.01 |
| | SGBM | 20.58 | 20.35 | 20.53 | 20.82 | 20.54 |
| 100 mm | RSBV | 20.26 | 20.33 | 20.27 | 19.79 | 20.26 |
| | SGBM | 20.83 | 20.68 | 20.32 | 20.68 | 20.64 |
| 150 mm | RSBV | 19.96 | 20.04 | 19.98 | 20.25 | 19.98 |
| | SGBM | 20.66 | 20.63 | 20.68 | 20.69 | 20.82 |

According to the feature point coordinates and the disparity values, the 3D coordinates of the target with the optical center of the left camera as the origin of the coordinate system could be obtained using the reprojection matrix. Figure 21 shows the scatter diagram drawn according to the *X* and *Y* coordinates reconstructed by the RSBV and SGBM algorithms. The depth value was normalized, and shapes of different sizes represent the difference between the depth values. The larger the shape is, the greater its depth value.

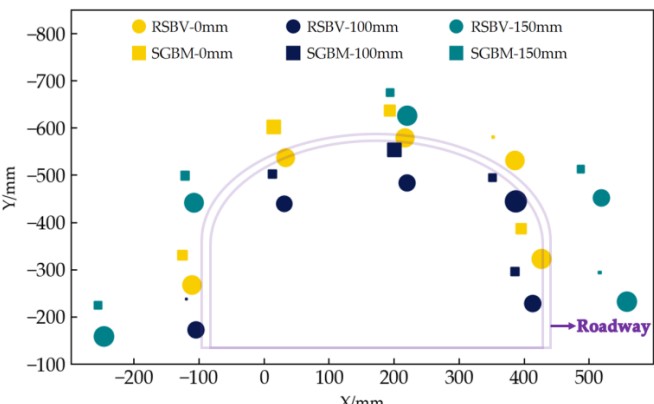

**Figure 21.** Three-dimensional coordinate comparison of target.

It can be seen from the figure that the main difference between the 3D coordinates reconstructed by the RSBV algorithm and the SGBM algorithm is the depth value. The depth value obtained by the RSBV algorithm is larger than that obtained by the SGBM algorithm, which is consistent with the disparity value. In the same algorithm, the difference between the depth values calculated by the RSBV algorithm is small, while the difference between the values calculated by the SGBM algorithm is large. The targets in this experiment were mainly arranged in the same plane. After moving the target, the depth value changed little, so the RSBV algorithm was more consistent with the actual experiment.

In order to verify the accuracy of the RSBV algorithm, the RSBV algorithm and the SGBM algorithm were used to carry out a three-dimensional reconstruction of the target feature points and obtain three-dimensional coordinates. Then, the relative distance between the two sides, the shoulder angles on both sides, and the adjacent target feature points were calculated by the distance formula and compared with the actual distance. The serial number of the target from left to right in the image is 1–5. The final measurement results of the two algorithms are shown in Table 4, and the measurement error in Figure 22 is plotted piecewise according to the actual distance. The bar chart describes the average absolute error, and the line chart describes the average relative error.

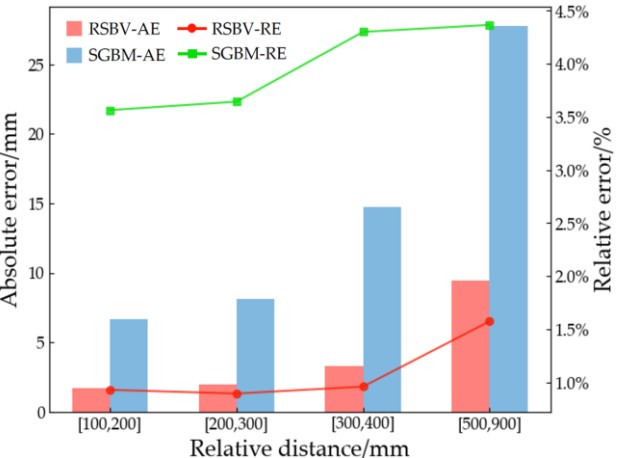

**Figure 22.** Measurement error.

**Table 4.** Comparison of deformation measurement results between SGBM algorithm and RSBV algorithm.

| Target Relative Distance | Moving Distance | Move 0 mm | | Move 100 mm | | Move 150 mm | |
|---|---|---|---|---|---|---|---|
| | Algorithm | RSBV | SGBM | RSBV | SGBM | RSBV | SGBM |
| 1–2 | Actual distance/mm | 304 | | 302 | | 320 | |
| | Measurement results/mm | 306.00 | 315.28 | 299.92 | 288.61 | 315.67 | 305.64 |
| | Absolute error/mm | 2.00 | 11.28 | −2.08 | −13.39 | −4.33 | −14.36 |
| | Relative error/% | 0.66% | 3.71% | −0.69% | −4.43% | −1.35% | −4.49% |
| 2–3 | Actual distance/mm | 190 | | 192 | | 381 | |
| | Measurement results/mm | 188.39 | 183.93 | 194.06 | 200.50 | 376.29 | 361.50 |
| | Absolute error/mm | −1.61 | −6.07 | 2.06 | 8.50 | −4.71 | −19.50 |
| | Relative error/% | −0.85% | −3.20% | 1.07% | 4.43% | −1.23% | −5.12% |
| 3–4 | Actual distance/mm | 177 | | 184 | | 344 | |
| | Measurement results/mm | 175.73 | 172.85 | 185.99 | 176.12 | 347.82 | 334.95 |
| | Absolute error/mm | −1.27 | −4.15 | 1.99 | −7.88 | 3.82 | −9.05 |
| | Relative error/% | −0.72% | −2.35% | 1.08% | −4.28% | 1.11% | −2.63% |
| 4–5 | Actual distance/mm | 215 | | 226 | | 228 | |
| | Measurement results/mm | 212.86 | 207.02 | 228.81 | 215.68 | 226.98 | 221.95 |
| | Absolute error/mm | −2.14 | −7.98 | 2.81 | −10.32 | −1.02 | −6.05 |
| | Relative error/% | −0.99% | −3.71% | 1.24% | −4.57% | −0.45% | −2.65% |
| 1–5 (Two sides) | Actual distance/mm | 554 | | 529 | | 815 | |
| | Measurement results/mm | 540.30 | 524.40 | 520.74 | 509.50 | 807.50 | 774.48 |
| | Absolute error/mm | −13.70 | −29.60 | −8.26 | −19.50 | −7.50 | −40.52 |
| | Relative error/% | −2.47% | −5.34% | −1.56% | −3.69% | −0.92% | −4.97% |
| 2–4 (Two shoulder angles) | Actual distance/mm | 350 | | 368 | | 619 | |
| | Measurement results/mm | 352.75 | 344.20 | 364.74 | 338.32 | 627.33 | 597.56 |
| | Absolute error/mm | 2.75 | −5.80 | −3.26 | −29.68 | 8.33 | −21.44 |
| | Relative error/% | 0.78% | −1.66% | −0.88% | −8.06% | 1.35% | −3.46% |

As seen from Table 4, the relative distance calculated by the RSBV method is close to the actual distance, and the measurement results fluctuate around the real value. In contrast, the measurement results of the SGBM algorithm are generally small and have significant errors. In addition, the average error of measurement results of the RSBV method is 4.1 mm, while the average error of the SGBM algorithm is 14.31 mm. The error of the RSBV algorithm is much smaller than that of the SGBM algorithm. As seen in Figure 22, the RSBV algorithm is significantly superior to the results of the SGBM algorithm in measuring the relative distance of targets within the same range. In different distance ranges, the RSBV algorithm measurement results are relatively stable, and the error changes are minor, especially when the relative distance is less than 400 mm. The relative error is less than 1%, maintaining high accuracy. When the relative distance is greater than 300 mm, the measurement error of the SGBM algorithm increases obviously. The average relative error measured by the RSBV algorithm is less than 1.6%, and the error variation is slight, while the relative error range of the SGBM algorithm is 3.5–4.4%. The error is generally significant and fluctuates wildly. In terms of computing speed, the RSBV method has obvious advantages. The average time for processing a set of binocular images is 1.87 s, while the average time of the SGBM algorithm is 8.95 s. The RSBV algorithm takes only 20% of the SGBM algorithm.

It can be seen from the above analysis that the RSBV algorithm has significant advantages over the SGBM algorithm in terms of accuracy and speed. Because the RSBV method adopts the SIFT feature matching method with epipolar constraints, and the coordinates of SIFT feature points were extracted as sub-pixel coordinates. Compared with SGBM, the target center coordinates were obtained more accurately by elliptic fitting, so the disparity value was calculated more accurately. In addition, SIFT feature points are sparsely matched, and only the detected target feature points need to be matched, which takes less time than the SGBM algorithm.

The above experimental results prove that the binocular vision algorithm can measure roadway deformation. The proposed real-time sensing method RSBV of deformation situation of mining roadway based on binocular stereo vision realized rapid and accurate automatic measurement of roadway host rock deformation in a specific range.

*5.5. Error Analysis*

According to the above error analysis, the roadway deformation calculated by the RSBV method is consistent with the actual value. The error is controlled within 1.6%, but there are still widespread errors. According to the analysis of the algorithm flow in the experiment, there are mainly three kinds of errors in the experiment.

1. Camera calibration error

In the process of camera calibration, the accuracy of the calibration board and algorithm plays a decisive role in the calibration results. The size of the black and white checkerboard of the calibration board is the reference value of the calculation of the world coordinate distance. Because the ordinary printer is not accurate enough, it causes an error in the calibration board and affects the final calibration result. The accuracy of checkerboard corner extraction also directly affects the positioning of world coordinates and thus affects the calibration results. In addition, the number of calibration images also causes errors. If the number of images is too small, the parameter calculation is inaccurate, and if the number of images is too large, the error accumulates. Therefore, the appropriate number of images is also the key factor for calibration.

2. Errors caused by equipment

The depth value is calculated with the focal length and baseline of the camera. The larger the focal length and baseline, the further the camera can measure. Because the camera used in this experiment is the fixed baseline and the focal length is small, there is an inevitable error in the disparity calculation of the target at a relatively long distance.

3. System error

In the process of 3D coordinate reconstruction, the calibration results of binocular cameras are used for calculation. Due to the calibration errors of cameras, errors in the 3D coordinate calculation are caused, which ultimately affect the deformation measurement results. In addition, various algorithms are used in the disparity calculation process, leading to error accumulation due to the algorithm's accuracy.

## 6. Conclusions and Discussion

An experimental platform for roadway deformation simulation was built by analyzing the deformation characteristics of mining roadways, and a roadway deformation situation perception method based on binocular stereo vision was built. The method was used for roadway deformation measurement experiment analysis, and the results showed that:

1. Within a distance of 3 m, the roadway deformation measurement error range by the RSBV method proposed in this study is less than 1.6%, and the average relative error is reduced by 2.88% compared with the measurement results based on the SGBM method. The average absolute error of the RSBV method is 4.11 mm, which is 10.2 mm lower than that of the SGBM method. The RSBV method takes 1.87 s to process a set of binocular images on average, which is only 20% of the SGBM algorithm, indicating that this paper's roadway deformation measurement method is more accurate and faster.

2. Through the experiment of extracting SIFT features from ROI images and the feature matching experiment of fusion epipolar constraint, the effectiveness of eliminating background interference image to reduce SIFT detection range and improve algorithm speed was revealed, as well as the effect of epipolar constraint method to improve feature-matching accuracy and speed.

3. The main innovation of this paper is that the binocular vision method is introduced into the deformation monitoring of the host rocks in the roadway under dim conditions, and considering the factors of low illumination and high dust in the actual environment, a series of methods for target image processing in the actual environment are proposed. Based on binocular vision, the roadway deformation monitoring test scene under dim conditions was designed, and the relative distance between monitoring targets was finally accurately obtained in real time. Through the experiment, the feasibility of continuous collection, accurate real-time measurement, and remote monitoring of the deformation of roadway host rocks using the binocular stereo vision method in the laboratory stage were verified.

In this study, the proposed roadway deformation monitoring method was tested by simulating the dark environment of the roadway. The experimental results show that the proposed method can accurately obtain the relative distance between targets in real time, and the errors meet the requirements of practical engineering. However, there are still some shortcomings.

- From the aspect of experimental design, the accuracy of depth measurement was limited because the binocular camera used in the experiment was a fixed baseline. The resolution of the camera was low, which made it challenging to meet the accurate recognition of the target image at a distance, so the experiment was only carried out at a distance of 3 m from the roadway.
- Because the quality of the image collected in the experiment was greatly affected by the light, it was necessary to provide a stable auxiliary light source for collecting and monitoring the target image.
- Due to the slow deformation of surrounding rock in the actual roadway environment and the high requirements for the safety of underground electronic equipment, this experiment has only been verified under laboratory conditions, not tested in the actual environment.

To sum up, given the above limitations, the future plan of our study is to take the light intensity and distance factors as variables and conduct experiments in the actual roadway environment to verify the effect of the RSBV method. In addition, the collected roadway deformation data can be further analyzed and utilized. Through the construction of a deep learning prediction model, it can be used to predict the deformation of the roadway surrounding rock and provide a reference for the roadway deformation trend. These data can further excavate the change of the roadway surrounding rock disturbed by the mining of the working face and provide a basis for roadway support.

**Author Contributions:** Conceptualization, methodology, and supervision, P.S. and X.L.; software, validation, visualization C.Y. and H.S.; formal analysis, X.C.; investigation and resources, C.L.; data curation, writing—original draft preparation, and writing—review and editing, C.Y.; funding acquisition, P.S. All authors have read and agreed to the published version of the manuscript.

**Funding:** This research was funded by National Natural Science Foundation of China, grant number 52274138, 51904227, U1965107; Innovation Capability Support Program of Shaanxi, grant number 2022KJXX-58; Yulin High-tech Zone Science and Technology Plan Project, grant number ZD-2021-01.

**Institutional Review Board Statement:** Not applicable.

**Informed Consent Statement:** Not applicable.

**Data Availability Statement:** Not applicable.

**Acknowledgments:** We thank the National Natural Science Foundation of China (52274138, 51904227, U1965107), Innovation Capability Support Program of Shaanxi (2022KJXX-58) and Yulin High-tech Zone Science and Technology Plan Project (ZD-2021-01) for its support of this study. We thank the academic editors and anonymous reviewers for their kind suggestions and valuable comments.

**Conflicts of Interest:** The authors declare no conflict of interest.

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
