# Peer review of "Evaluation of Real-Time Perception of Deformation State of Host Rocks in Coal Mine Roadways in Dusty Environment"

_sustainability, doi:10.3390/su15032816_

Round 1
Reviewer 1 Report
The present manuscript proposes a real-time deformation situation sensing method based on binocular stereo vision for dynamic and accurate real-time acquisition of relative deformation of the surrounding rock in a dark environment.
It is a very interesting research paper, carefully written and based on a well-conducted research campaign.
The subject is well explained, and research is conducted with valid arguments.
It has an extensive and current bibliographic collection. It also has a very consistent discussion and analysis to compare experimental results with other authors.
It is well formatted and composed with the necessary, sufficient and self-explanatory images.
just a fix:
Write the abbreviation RSBV in full the first time it appears in the text
Reviewer 2 Report
In this manuscript the authors presented “experimental study for real-time Perception of Deformation state of Surrounding Rock in Mining Roadway under Dusky Environment”. In my point of view, it must be revised based on the following comments:
1) The language must be checked and corrected by a native English speaker.
2) The title can be changed to “Evaluation of Real-time Perception of Deformation state of host Rocks in Coal Mines Roadway under Dusky Environment”.
3) The abstract is too long hence, it must be revised.
4) Some information can be obtained below paper regarding detection of convergency in tunnel. https://doi.org/10.1007/s40534-018-0173-y
5) The authors must present the location of case study and geotechnical properties of host rocks in the manuscript.
6) What is anchor spray in the line 123? Are aim of the authors rockbolt or anchor installation?
7) Line 127, instead of ‘dotted line’ the word “dashed line’ is better.
8) Some words such as “in this paper’ or ‘this paper’ have been repeated too many hence, the whole paper must be revised. Because using these words inside the main body of manuscript isn’t common.
9) What is ROI? It must be illustrated in the text.
10) How have been verified the results? did the authors check its result with real condition? It must be described in the text.
Reviewer 3 Report
The present manuscript describes the investigation into solving problems of monitoring technology of surrounding rock deformation in the current of coal mines, such as large error, information lag, and low frequency of mining analysis, a real-time perception method of deformation situation based on binocular stereo vision is proposed, which realizes dynamic, accurate real-time acquisition of surrounding rock relative deformation in a dusky environment. A manuscript has a practical application and also provides important theoretical for the next studies. The paper can be accepted for publication after providing the corrections mentioned below.
Issue 1. The abstract section sounds unclear. It must be more conscious. Even the volume of the abstract must be decreased in twice. Please refer to the guide for authors prepared by MDPI to see how to prepare an abstract.
Issue 2. In the Introduction section, an enhanced literature review is required. For this study, the authors have used only 16 literature sources. It seems insufficient for such type of research. Moreover, almost references come from China.
Issue 3. It will be great if the authors show some description in context – Why it is important to conduct this study? What is limitations?
Issue 4. I would loke to suggest considering two research in your paper.
I. It is well known that underground monitoring is the best way of roadways support design validation in a long time period. Please refer to the paper below. Małkowski, P., Niedbalski, Z., Majcherczyk, T., & Bednarek, Ł. (2020). Underground monitoring as the best way of roadways support design validation in a long time period. Mining of Mineral Deposits, 14(3), 1-14. https://doi.org/10.33271/mining14.03.001
This paper shows the importance of geotechnical monitoring in assessing stability of an underground excavation. Every mining excavation is designed on the basis of limited geotechnical data and with some physical assumptions.
II. Sakhno, I., Liashok, Ia., Sakhno, S., & Isaienkov, O. (2022). Method for controlling the floor heave in mine roadways of underground coal mines. Mining of Mineral Deposits, 16(4), 1-10. https://doi.org/10.33271/mining16.04.001
Controlling the floor heave of mine roadways located in the zone of increased stresses is a very important issue. Please accent your attention on it.
Issue 5. The aim and the tasks must be highlighted at the end of the introduction section.
Issue 6. Figure 10. SIFT features descriptor must be discussed in more detail.
Issue 7. I am not sure that it is necessary to give Figure 12.
Issue 8. Why it is important to give Binocular camera calibration results that is described on Table 1?
Issue 9. A short description of further research must be given in the discussion section.
Issue 10. What is the novelty of the research? The novelty of the paper must be highlighted in the conclusions section.
Issue 11. I must admit that a very good study was performed, and I will recommend your paper for publication after step by step careful revision.
Reviewer 4 Report
This article is really interesting. The author proposed an RSBV method based on binocular stereo vision and built a roadway surrounding the rock deformation test scene under dark conditions for practical verification. In particular, the processing process of monitoring the target image was deeply studied, and good results were obtained. In summary, the whole article is a scientific paper with clear logic, prominent innovation points, illustrations and text, and complete analysis.
Suggest to accept, but some problems need to be solved.
Comment 1: Experiments verify the advantages of the RSBV algorithm in this paper by comparing it with the global stereo matching algorithm SGBM, and why the RSBV algorithm is superior to the SGBM method in accuracy and speed.
Comment 2: The establishment of a roadway coordinate system is particularly important. The transformation relationship between different coordinate systems is related to the accuracy of deformation data. How to do it in this paper?
Comment 3: It is suggested that the introduction should be more detailed, and more references can be cited to further elaborate the author's literature.
Comment 4: In the introduction, the following two references are recommended to support the sentence of "surrounding rock has the potential safety hazard of instability and destruction". (10.1007/s00603-022-03160-8, 10.1186/s40069-022-00547-3)
Round 2
Reviewer 2 Report
all my comments have been considered in the revised paper hence, it is acceptable.
Reviewer 3 Report
Dear authors, I am more than satisfied with the corrections provided by you.
This study is an important contribution to sustainable mining.
Congratulations to the authors.